# Treatment of Domestic Wastewater in Small-Scale Sand Filters Fortified with Gypsum, Biotite, and Peat

Kati Martikainen *, Anna-Maria Veijalainen, Eila Torvinen  and Helvi Heinonen-Tanski

Department of Environmental and Biological Sciences, University of Eastern Finland, P.O. Box 1627, FI-70211 Kuopio, Finland
* Correspondence: kati.a.martikainen@uef.fi

**Abstract:** Sand filtration is a low-cost and easy solution for household wastewater treatment in areas lacking a centralized sewage system. However, there are only a few studies about the treatment efficiencies of nutrients and enteric microorganisms and their removal or filter mass reuse potential. Sand columns with different phosphorus adsorbents, gypsum, biotite, and peat were tested in laboratory-scale filters at 4 °C to assess their performance in variable conditions and their possibility to increase the efficiency of sand filters. The columns were fed with real municipal wastewater with variable wastewater flow and phosphate load at different stages of the experiments. Phosphate and total nitrogen concentrations were low in the effluent of all columns, and they were mostly rather similar. Waste gypsum was found to greatly increase the conductivity of the effluent. The numbers of enteric microorganisms in the effluents were low, and the used filter masses achieved good hygienic quality after the tests. Phosphate, ammonium, and nitrate concentrations were low in the used masses, evidently since the columns had operated only for 21–30 weeks. Sand filtration proved to be an effective method for wastewater treatment, but changing conditions should be considered when designing these filters. The masses have reuse potential as soil improvement.

**Keywords:** column experiment; domestic wastewater; irrigation water; phosphorus; sand filter; total nitrogen



## 1. Introduction

Soil Treatment Systems (STSs), which consist of septic tanks and a sand filter, are used worldwide in single households and small communities. They are suitable especially if the connection to sewer pipes is economically not feasible or is impossible due to long distances or difficult terrain for building the sewer network. STSs could also be a good choice in developing areas because of their low costs, easy use, and ability to function without electricity [1–6]. However, the capacity of sand filters to bind phosphorus (P) may sometimes be insufficient [7,8] which can cause risks to the quality of nearby natural surface waters, because P is often a limiting nutrient [9–11]. The discharge of limiting nutrients is known to accelerate eutrophication, causing anoxic bottom zones and algal blooms in sea areas, such as the Baltic Sea [12], and fresh waters [13]. In addition, microbial contamination of STSs' effluents can be a concern [14,15] and lead to health risks via contaminated groundwaters or surface waters used for irrigation or recreational purposes [16–19].

The removal of phosphorus from domestic wastewater has been studied by using the STSs with layers of different natural materials such as bentonite and sepiolite clays, apatite, wollastonite, shell sand, and gravel [20–25]; different industrial by-products such as biotite, water-cooled blast furnace slag, and oil shale ash [21,26,27]; or materials prepared for this purpose such as Polonite®, Filtra P and Filtralite P®, Rockfos®, Lega® [21,23,28]. Many sand filter studies have been conducted in the laboratory or at a pilot scale with synthetic wastewater or nutrient solutions [21,22,24] and the results for phosphorus removal have been varying. Only a few full-scale studies have shown that P removal can be improved

if adsorbing material is used in constructed wetlands [25,29] or sand filters [27,30,31], and, therefore, more research is needed. It is necessary to determine whether the studied materials are suitable to adapt to different wastewater flow and nutrients loads. It would be ideal to find an efficient and low-cost method that would apply for each local situation [6].

Rock phosphate is an essential source of P in mineral fertilizers, but, unfortunately, this resource is estimated to end within a couple of hundred years, and, therefore, P recovery is important [2,11,32,33]. P-producing countries may restrict their exports before reserves are depleted [34], hence the need for recycling P increases. Adsorbing P from wastewater with sand filters offers a possibility to reuse the filter masses and recycle P if the quality of the masses is high enough [21,35,36]. For example, Polonite®, Filtralite P, and wollastonite have been found to be suitable materials for adsorbing P from wastewater [21], and, thus, the filters could be reused as soil amendments improving soil fertility [7,36,37]. Masses must not pose a secondary risk of contamination [38]. If the mass meets these requirements but is not sufficiently functional as an adsorbent, modification often improves its performance [38]. A Ca-based adsorbent, waste gypsum, is a by-product of the phosphoric acid industry. It has been successfully used in the Finnish coastal area to reduce agricultural P runoff [39], and it might be an option for binding P during wastewater soil filtration. The reuse of by-product would be beneficial also because it would decrease the costs caused by the disposal or storage of it, and, thus, it is worth studying here. Moreover, biotite is a by-product of the phosphorus acid industry. The results show that it works for phosphorus removal, but there is variation in its performance [14,15,27]. Peat resources are rather rich in Finland, and peat has a high absorption capacity of nutrients from manure [40]. It has also been used successfully in a few P removal studies on wastewater [41–44], and, therefore, the potential of peat is worth studying here.

The aim of the study was to investigate whether P-absorbing layers of industrial by-products such as biotite and waste gypsum mixed with sand or sand and peat in STS could improve the wastewater treatment of sand filters. The experiments were conducted with different loading rates of real wastewater to find out the performance in varying circumstances of households. The used filter masses were analyzed for their hygienic and chemical qualities to estimate their reuse potential in plant production.

## 2. Materials and Methods

### 2.1. Column Experiments and Wastewater Sampling

Two filtration experiments were carried out in columns to simulate sand filters typically used in rural areas. Polyvinyl Chloride (PVC) tubes (height of 130 cm and inner diameter of 10 cm) were filled with a 15 cm layer of gravel (ø = 2–4 cm), a 30 cm layer of filter sand (ø = 0–8 mm), a 30 cm layer of P-binding material, and, on top, another 30 cm layer of filter sand (Figure 1). P-binding layers were a mixture of filter sand plus waste gypsum (column G) (2:3, *v/v*) and biotite (column B) (30 cm layer biotite) in experiment (Exp.) 1 and waste gypsum plus filter peat (column GP) (1:1, *v/v*) and waste gypsum plus filter sand and filter peat (column GSP) (2:1:1, *v/v/v*) in Exp. 2. Waste gypsum was mixed with sand and/or peat to get better permeability to avoid a blocking effect. In the control sand (column S) (one in both experiments), there was only filter sand without a P-binding layer. All columns were gradually filled and compacted gently throughout the filling. After Exp. 1, the columns were cleaned and filled with new materials for Exp. 2. as shown in Figure 1. The columns were watered with 5–10 L of tap water per column before starting the experiments. The experiments were started after the water had stopped running through them. Biotite and waste gypsum are by-products from the apatite mine (Yara, Finland) from Siilinjärvi Finland. The sand was from Rudus Oy from Kuopio, Finland.

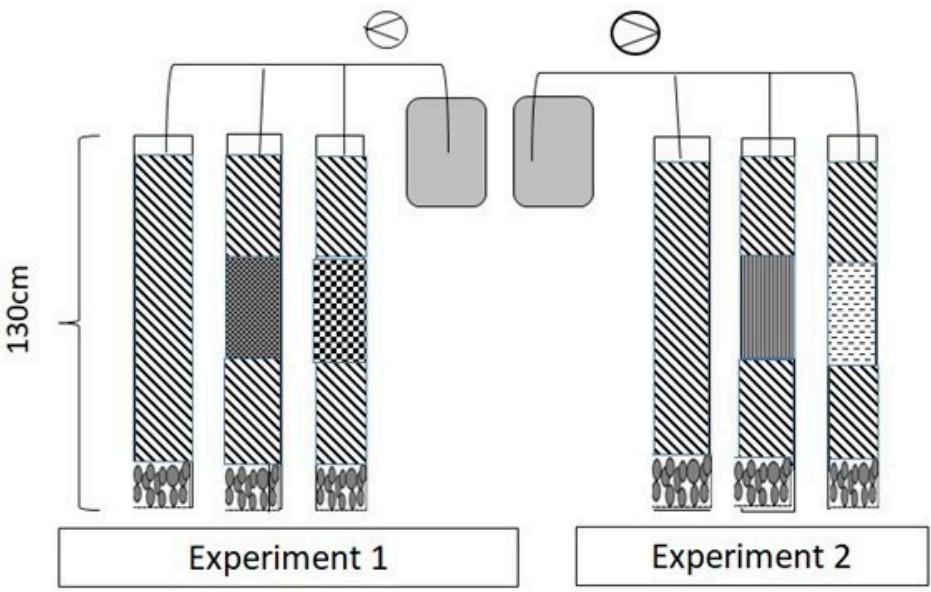

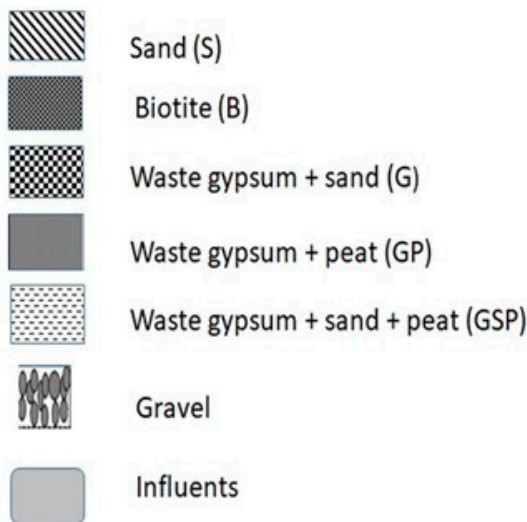

Sand (S)

Biotite (B)

Waste gypsum + sand (G)

Waste gypsum + peat (GP)

Waste gypsum + sand + peat (GSP)

Gravel

Influents

**Figure 1.** Filter columns and setups in Experiments 1 and 2; sampling from the gravel layer.

Screened municipal influent collected every 7–10 days (Lehtoniemi WWTP, Kuopio, Finland) was pumped to the top of the columns twice a day with three different total loading rates: 42 L m$^{-2}$ as a normal loading (330 mL day$^{-1}$), 21 L m$^{-2}$ as a low loading (165 mL day$^{-1}$), and 52 L m$^{-2}$ as a high loading (410 mL day$^{-1}$). The experiments began with a normal loading stage (NL) (3 weeks), and they were followed by consecutive stages of high loading (3 weeks), NL (3 weeks), low loading (4 weeks in Exp. 1 and 3 weeks in Exp. 2.), NL (3 weeks), NL with extra P (3 weeks), and, finally, NL (11 weeks in Exp. 1. and 4 weeks in Exp. 2.). Phosphorus was added as $KH_2PO_4$ (Merck KGA Germany)—in Exp. 1 as five times the normal $PO_4$-P concentration to reach appr. 50 mg P L$^{-1}$ (Table 1a) and in Exp. 2 as two times to reach the concentration of appr. 7 mg P L$^{-1}$ (Table 1b). The experiments were carried out at +4 °C to simulate conditions in a cold climate. The water that flowed through the columns (effluent) were collected in bottles. Roadmap is presented in Figure 2.

**Table 1.** (**a**) Exp. 1 with the columns sand (S), biotite (B), and gypsum (G). $PO_4$-P, $COD_{Cr}$, and $N_{tot}$ concentrations are presented as averages $\pm$ standard deviations (sd) and reduction-% (red%) as average (min–max) in influent and effluents of the columns. No red = no reduction found. Statistically significant differences between the columns S and B and between the columns S and G at different loading stages are presented as * = $p < 0.05$, ** = $p < 0.01$, *** = $p < 0.001$. (**b**) Exp. 2 with the columns sand (S); gypsum and peat (GP); and gypsum, sand, and peat (GSP). $PO_4$-P, $COD_{Cr}$, and $N_{tot}$ concentrations are presented as averages $\pm$ sd and reduction-% as average (min–max) in influent and effluents of the columns. No red = no reduction found. Statistically significant differences between the columns sand (S) and GP and between the columns S and GSP at different loading stages are presented as * = $p < 0.05$, ** = $p < 0.01$.

| (a) Exp. 1 | | | | | |
|---|---|---|---|---|---|
| **Wastewater Loading and Its Duration** | **Parameter** | **Influent** | **Effluents** | | |
| | | | **Column S** | **Column B** | **Column G** |
| Normal loading 42 L m$^{-2}$ (3 weeks) | $PO_4$-P mg L$^{-1}$ | 10 ± 3.1 | 0.05 ± 0.03 | 0.3 ± 0.01 | 0.05 ± 0.02 |
| | red% $PO_4$-P | | 99.6 (99.3–99.9) | 99.3 (99.6–99.9) | 99.4 (98.8–99.8) |
| | $N_{tot}$ mg L$^{-1}$ | 78 ± 7.6 | 25 ± 21 | 91 ± 13 | 82 ± 0.21 |
| | red% $N_{tot}$ | | 65.3 (43.0–87.6) | no red | no red |
| | COD mg L$^{-1}$ | 540 ± 270 | 34 ± 9.3 | 32 ± 5.7 | 77 ± 65 |
| | red% COD | | 93.3 (92.5–94.7) | 93.6 (92.1–95.5) | 87 (82.1–90.8) |
| High loading 52 L m$^{-2}$ (3 weeks) | $PO_4$-P mg L$^{-1}$ | 7 ± 0.1 | 0.01 ± 0.02 | 0.02 ± 0.03 | 0.07 ± 0.03 * |
| | red% $PO_4$-P | | 99.8 (99.1–99.9) | 99.7 (98.9–99.9) | 99.1 (97.8–99.3) * |
| | $N_{tot}$ mg L$^{-1}$ | 170 ± 156 | 27 ± 23 | 107 ± 11 * | 83 ± 2.2 |
| | red% $N_{tot}$ | | 83.5 (81.5–84.6) | 5.5 (no red-65.7) | 24.5 (no red-75.7) |
| | COD mg L$^{-1}$ | 470 ± 27 | 44 ± 7.2 | 44 ± 5.1 | 40 ± 7.8 |
| | red% COD | | 90.8 (89.7–91.6) | 90.7 (90.1–91.1) | 94.5 (91.1–92.4) |
| Normal loading 42 L m$^{-2}$ (3 weeks) | $PO_4$-P mg L$^{-1}$ | 7 ± 0.3 | 0.03 ± 0.03 | 0.02 ± 0.01 | 0.18 ± 0.19 ** |
| | red% $PO_4$-P | | 99.6 (98.8–99.9) | 99.8 (99.5–99.9) | 97.3 (92.3–99.3) * |
| | $N_{tot}$ mg L$^{-1}$ | 370 ± 27 | 72 ± 20 | 106 ± 11 | 70 ± 14 |
| | red% $N_{tot}$ | | 80.5 (74.3–85.8) | 71.4 (70.3–72.5) | 81.1 (79–83.7) |
| | COD mg L$^{-1}$ | 490 ± 9.6 | 50 ± 0.6 | 51 ± 3.0 | 37 ± 2.1 |
| | red% COD | | 89.7 (89.4–90) | 89.5 (88.8–90.1) | 92.3 (92.1–92.8) |
| Low loading 21 L m$^{-2}$ (4 weeks) | $PO_4$-P mg L$^{-1}$ | 6.1 ± 1.7 | 0.03 ± 0.02 | 0.02 ± 0.01 | 0.26 ± 0.18 *** |
| | red% $PO_4$-P | | 99.6 (98.9–99.9) | 99.7 (99.5–99.9) | 95.8 (88.6–98.2) *** |
| | $N_{tot}$ mg L$^{-1}$ | 260 ± 82 | 70 ± 12 | 88 ± 2.1 | 71 ± 17 |
| | red% $N_{tot}$ | | 70.9 (59.8–82.4) | 69.8 (65.6–74.0) | 70.9 (61.0–82.4) |
| | COD mg L$^{-1}$ | 385 ± 217 | 48 ± 15 | 60 ± 27 | 32 ± 12 |
| | red% COD | | 80.8 (56.7–89.4) | 51.3 (no red-91.6) | 87.6 (73.3–93.6) |
| Normal loading 42 L m$^{-2}$ (3 weeks) | $PO_4$-P mg L$^{-1}$ | 7.4 ± 5.1 | 0.02 ± 0.03 | 0.02 ± 0.01 | 0.33 ± 0.16 *** |
| | red% $PO_4$-P | | 99.7 (99.5–99.9) | 99.6 (98.3–99.9) | 94.6 (86.7–98.2) *** |
| | $N_{tot}$ mg L$^{-1}$ | 35 ± 6.1 | 46 ± 3.3 | 52 ± 1.9 | 34 ± 8.1 |
| | red% $N_{tot}$ | | no red | no red | 4.8 (no red -11) |
| | COD mg L$^{-1}$ | 270 ± 260 | 54 ± 8.0 | 12 ± 6.0 | 47.7 ± 8.5 |
| | red% COD | | 65.6 (50.8–91.8) | 91.0 (85.7–98.9) | 72.5 (61.9–90) |
| $PO_4$-P adding with normal loading 42 L m$^{-2}$ (3 weeks) | $PO_4$-P mg L$^{-1}$ | 46 ± 19 | 0.05 ± 0.06 | 0.03 ± 0.03 | 2.51 ± 3.05 *** |
| | red% $PO_4$-P | | 99.9 (99.2–99.9) | 99.9 (99.4–99.9) | 91.3 (66.7–99.5) *** |
| | $N_{tot}$ mg L$^{-1}$ | 75 ± 11 | 44 ± 1.7 | 46 ± 2.9 | 46 ± 5.1 |
| | red% $N_{tot}$ | | 40.6 (33.3–48.6) | 37.9 (34.7–43.4) | 38.5 (36.4–40.7) |
| | COD mg L$^{-1}$ | 390 ± 270 | 87 ± 19 | 52 ± 17 | 83 ± 29 |
| | red% COD | | 69.5 (56.1–89.5) | 84.1 (78.3–89.8) | 69.2 (53.3–92.3) |
| Normal loading 42 L m$^{-2}$ (11 weeks) | $PO_4$-P mg L$^{-1}$ | 10 ± 5.2 | 0.24 ± 0.22 | 1.51 ± 1.07 ** | 6.21 ± 3.42 *** |
| | red% $PO_4$-P | | 97.4 (92.9–99.9) | 83.6 (64.3–99.5) ** | 31.3 (no red-86.3) *** |
| | $N_{tot}$ mg L$^{-1}$ | 90 ± 5.5 | 51 ± 3.6 | 47 ± 5.9 | 51 ± 3.0 |
| | red% $N_{tot}$ | | 43.4 (40.9–50.4) | 47.7 (40–51.4) | 43.6 (43.2–44.1) |
| | COD mg L$^{-1}$ | 350 ± 36 | 130 ± 7.8 | 44 ± 8.6 * | 90 ± 14 |
| | red% COD | | 62.2 (57.5–68.6) | 87.7 (84.7–88.8) ** | 73.9 (69.6–83.6) |

**Table 1.** *Cont.*

| Wastewater Loading and Its Duration | Parameter | Influent | Effluents | | |
|---|---|---|---|---|---|
| | | | Column S | Column GP | Column GSP |
| Normal loading 42 L m$^{-2}$ (3 weeks) | PO$_4$-P mg L$^{-1}$ | 6.8 ± 0.6 | 0.07 ± 0.03 | 0.10 ± 0.08 | 0.15 ± 0.09 |
| | red% PO$_4$-P | | 98.8 (98.3–99.4) | 98.5 (96.6–99.8) | 97.7 (95.4–99.3) |
| | N$_{tot}$ mg L$^{-1}$ | 104 ± 7.2 | 31 ± 2.6 | 28 ± 13 | 27 ± 0 |
| | red% N$_{tot}$ | | 70 (67–75) | 73.9 (61.8–81) | 74.5 (73–76) |
| | COD mg L$^{-1}$ | 240 ± 7.6 | 79 ± 3.6 | 110 ± 50 | 110 ± 0 |
| | red% COD | | 67.3 (65–70) | 55 (40–77) | 54.2 (52.8–55.6) |
| High loading 52 L m$^{-2}$ (3 weeks) | PO$_4$-P mg L$^{-1}$ | 8.0 ± 0.3 | 0.04 ± 0.01 | 0.04 ± 0.01 | 0.03 ± 0.01 |
| | red% PO$_4$-P | | 99.5 (99.3–99.7) | 99.5 (99.3–99.7) | 99.6 (96.9–99.2) |
| | N$_{tot}$ mg L$^{-1}$ | 150 ± 16 | 52 ± 19 | 59 ± 24 | 64 ± 6.9 |
| | red% N$_{tot}$ | | 65.3 (58–76.9) | 60.6 (46–74.6) | 56.3 (52–63) |
| | COD mg L$^{-1}$ | 160 ± 54 | 91 ± 8.1 | 50 ± 14 * | 69 ± 2.0 |
| | red% COD | | 37.9 (14.7–62.6) | 67.1 (52–77.5) | 53.9 (40.4–67.8) |
| Normal loading 42 L m$^{-2}$ (3 weeks) | PO$_4$-P mg L$^{-1}$ | 7.2 ± 0.8 | 0.06 ± 0.02 | 0.14 ± 0.06 * | 0.05 ± 0.03 |
| | red% PO$_4$-P | | 99.1 (98.8–99.4) | 98.1 (96.9–99.2) * | 99.3 (98.7–99.7) |
| | N$_{tot}$ mg L$^{-1}$ | 92 ± 38 | 36 ± 15 | 58 ± 26 | 53 ± 23 |
| | red% N$_{tot}$ | | 60.4 (58–64) | 38.1 (35.2–42) * | 42.8 (40–46) |
| | COD mg L$^{-1}$ | 700 ± 220 | 59 ± 24 | 97 ± 68 | 83 ± 25 |
| | red% COD | | 91.4 (88.4–94.7) | 87.1 (81.3–90.9) | 88.1 (86.6–89.6) |
| Low loading 21 L m$^{-2}$ (3 weeks) | PO$_4$-P mg L$^{-1}$ | 8.3 ± 1.0 | 0.05 ± 0.02 | 0.08 ± 0.03 | 0.05 ± 0.03 |
| | red% PO$_4$-P | | 99.4 (99.1–99.7) | 99.0 (98.4–99.7) | 99.4 (98.9–99.8) |
| | N$_{tot}$ mg L$^{-1}$ | 150 ± 1.2 | 62 ± 1.7 | 46 ± 41 | 78 ± 0.0 |
| | red% N$_{tot}$ | | 58.5 (58–59.5) | 36 (0–60) | 48 (48) |
| | COD mg L$^{-1}$ | 200 ± 46 | 69 ± 24 | 130 ± 46 | 100 ± 39 |
| | red% COD | | 63.4 (53.3–77.2) | 34.1 (29.4–40.4) | 45 (31.7–63.2) |
| Normal loading 42 L m$^{-2}$ (3 weeks) | PO$_4$-P mg L$^{-1}$ | 8.3 ± 0.9 | 0.12 ± 0.03 | 0.06 ± 0.02 | 0.12 ± 0.09 |
| | red% PO$_4$-P | | 99.4 (99.1–99.7) | 99.3 (98.9–99.7) * | 98.6 (97.1–99.5) |
| | N$_{tot}$ mg L$^{-1}$ | 160 ± 13 | 67 ± 6.4 | 115 ± 45 | 88 ± 17 |
| | red% N$_{tot}$ | | 58.5 (53.5–64) | 44 (-2.5–84.4) | 44.6 (28–53.7) |
| | COD mg L$^{-1}$ | 550 ± 250 | 130 ± 56 | 150 ± 20 | 110 ± 10 |
| | red% COD | | 75.8 (53.3–76.5) | 65.5 (29.4–80.6) | 76.7 (31.7–83.5) |
| PO4-P adding with normal loading 42 L m$^{-2}$ (3 weeks) | PO$_4$-P mg L$^{-1}$ | 16 ± 0.7 | 0.25 ± 0.04 | 0.18 ± 0.08 | 0.20 ± 0.06 |
| | red% PO$_4$-P | | 98.5 (98.2–99.1) | 98.9 (98.2–99.1) | 98.8 (98.1–99.2) |
| | N$_{tot}$ mg L$^{-1}$ | 120 ± 3.4 | 36 ± 4.9 | 54 ± 10 * | 49 ± 9.2 |
| | red% N$_{tot}$ | | 69.9 (65.8–72.5) | 54.8 (46.1–59.2) | 58.7 (51–63.5) |
| | COD mg L$^{-1}$ | 280 ± 5.5 | 115 ± 52 | 170 ± 9.0 | 115 ± 4.4 |
| | red% COD | | 58.1 (35.2–70.2) | 38.1 (35.2–43.1) | 58.2 (57.3–59.3) |
| Normal loading 42 L m$^{-2}$ (3 weeks) | PO$_4$-P mg L$^{-1}$ | 6.9 ± 0.3 | 0.02 ± 0.02 | 0.19 ± 0.02 ** | 0.08 ± 0.03 |
| | red% PO$_4$-P | | 98.7 (98.2–99.1) | 97.3 (96.9–97.8) ** | 98.8 (98–99.3) |
| | N$_{tot}$ mg L$^{-1}$ | 100 ± 35 | 17 ± 15 | 27 ± 10 | 34 ± 12 |
| | red% N$_{tot}$ | | 85.8 (73.3–96.8) | 72 (64.8–82.5) | 66.7 (65–68) |
| | COD mg L$^{-1}$ | 270 ± 18 | 83 ± 31 | 130 ± 61 | 170 ± 58 |
| | red% COD | | 69.4 (57.5–77.2) | 53.9 (37.3–77.2) | 53.9 (6.4–57.9) |

*(b) Exp. 2*

### 2.2. Physico-Chemical Analyses of Influent and Effluent

Phosphate (PO$_4$) was analyzed with molybdate and ascorbic acid method (Hach DR 2010 or 2800, Method 8048) 3–5 times per week, and total nitrogen (N$_{tot}$) with persulfate method (Hach DR 2010, Method 10071) once a week. Chemical oxygen demand (COD$_{Cr}$) was determined with digestion method (Hach DR 2010 Method 8000) once a week. The pH and electrical conductivity were measured with a Hach Hqd Portable meter (Hach Co., Loveland, CO, USA) 3–5 times a week.

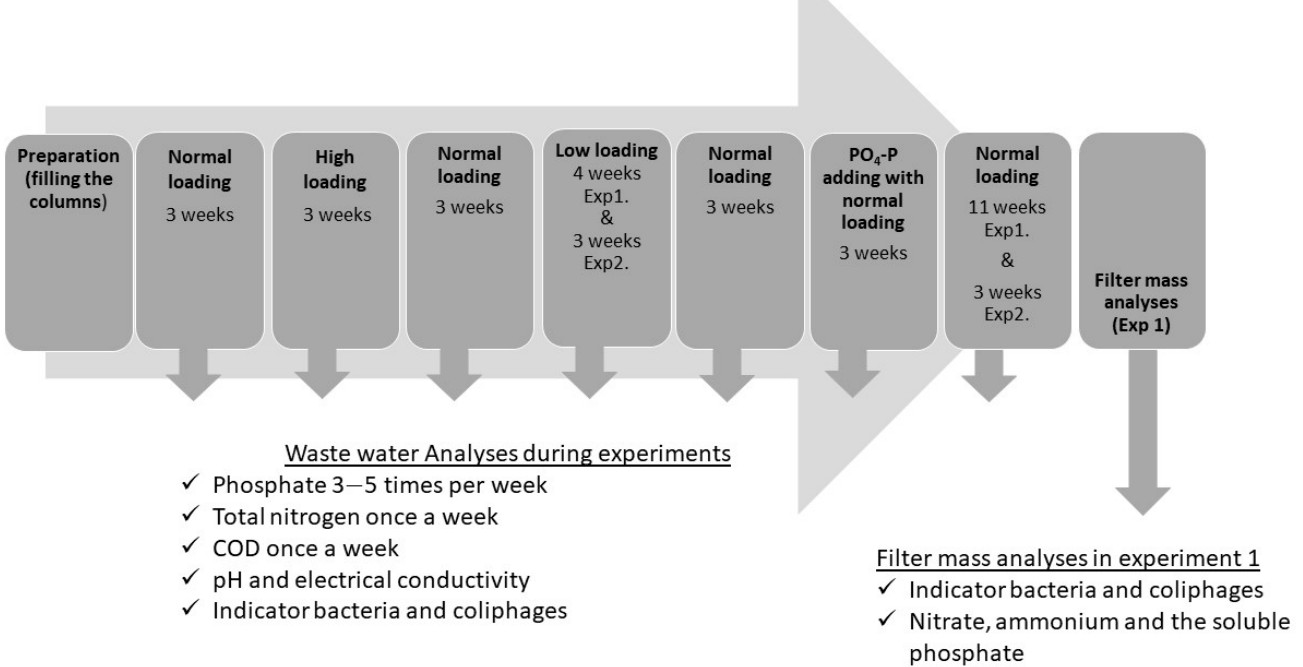

**Figure 2.** Research roadmap.

## 2.3. Bacteriological and Virus Analyses of Influent and Effluent

Fecal coliforms, *Escherichia coli*, sulfite-reducing clostridia, intestinal enterococci, and somatic and F-specific coliphages (hosts ATCC 13706 and ATCC 15597, respectively) were determined from the influent and effluent samples once a week according to the standard methods as previously described [13]. Kovac's indole reagent was used for confirmation of *E. coli* in Exp. 2. The detection limits were 1 or 10 CFU 100 mL$^{-1}$ for the filtration method and 5 CFU 1 mL$^{-1}$ for the spread plate method.

## 2.4. Filter Mass Analyses

After Exp. 1, the columns S, G, and B were divided into six sublayers with 15 cm lengths, and three sub-samples were collected from each sublayer for the microbiological and chemical analyses. The final results were calculated as a geometric mean of the six analyses of the top, middle, and bottom layers of the column. For the analyses of indicator microbes, 10 g of sample was diluted in 90 mL of sterile deionized water and shaken for 15 min in a gyratory shaker. *E. coli* (confirmed with Kovac's reagent), sulfite-reducing clostridia, intestinal enterococci, and F-specific coliphages (host ATCC 15597) were determined from liquid phase with the methods as described above for the water samples.

Nitrate ($NO_3^-$) concentrations were analyzed from water extracts with an ion chromatograph (DX 120, Dionex Corporation, USA), and ammonium ($NH_4^+$) concentrations were measured from 1 M KCl extracts with a spectrophotometer (Ultrospec 3000 Pro, Biochrom, UK) [45]. Total nitrogen concentration was calculated as the sum of ammonium– and nitrate–nitrogen concentrations. The soluble phosphate was extracted from the air-dried samples [46] with acid ammonium acetate, and the phosphate was determined with the Finnish standard SFS-1189 [47] which corresponds with the American standards [48] measuring the formed blue antimony phosphomolybdate complex at 880 nm (UV-2401 PC, UV-Vis recording spectrophotometer, Shimadzu, Japan). For the calculation of the results, dry weights of the soil samples were determined according to the method described [49]. Chemical parameters were analyzed as three replicates.

*2.5. Data Analyses*

The averages and standard deviations (sd) of parameters were calculated, and the statistical analyses of Exp. 1 and 2 were performed separately because different influents were used in the experiments. Half of the detection limit of microbes was used in the statistical analyses if the result was below the detection limit. The statistical differences in the reductions and concentrations between the control sand filter and filters with adsorbents during different wastewater loading were analyzed using the Kruskal–Wallis test with pair-wise comparison. The statistical difference in the concentrations of the used filter masses between the control sand filter and filters with adsorbents were also analyzed using the Kruskal–Wallis test with pair-wise comparison. Statistical analyses were conducted using IBM SPSS statistics 21 and 23.

## 3. Results

*3.1. Phosphate-P*

The $PO_4$-P concentrations of influent varied during Exp. 1 from 2.9 to 78 mg $L^{-1}$ and in Exp. 2 from 6.3 to 17 mg $L^{-1}$. The highest concentrations were measured at the $PO_4$-P adding stage during normal loading; average concentrations were 46 and 16 mg $L^{-1}$ in Exp. 1 and 2, respectively (Table 1a,b).

In Exp. 1, the concentrations of $PO_4$-P in all effluents from columns S, B, and G were usually low, and the average reductions were more than 90% (Table 1a), which is clearly above the required 70% for the reduction of total P [50], with a minimum of 0.01, 0.02, and 0.05 mg $L^{-1}$ average concentrations in the effluents of filters S, B, and G, respectively. Only at the last stage, which was the normal loading after the $PO_4$-P adding stage with the normal loading, the P reductions of the columns B ($p < 0.01$) (average reduction 83.6%) and G ($p < 0.001$) (average reduction 31.3%) were significantly lower than that of column S (average reduction 97.4), with the average concentrations of 0.24, 1.51, and 6.21 mg $L^{-1}$ in the S, B, and G columns, respectively. The reduction of column B was sporadic and decreased to 64.3 % at the lowest, while no reduction at all was noticed in column G for about a week. The decrease in the reduction efficiency of both columns improved towards the end of the experiment, but earlier in column B than in column G. In the last week, week 11, the reductions reached 97.9, 82.8, and 79.3% in the S, B, and G columns, respectively.

Similarly, in Exp. 2, the concentrations of $PO_4$-P in all effluents from all tested columns (S, GP, and GSP) were as low as 0.02, 0.04 and 0.03 mg $L^{-1}$ on average in S, B, and G columns, respectively, and the reductions were above 90% during the whole experiment, as shown in Table 1b. Only the column GP, after the high loading stage ($p < 0.05$) and after normal loading following the $PO_4$-P adding stage with the normal loading ($p < 0.01$), was less inefficient than the control S column, but the reduction in these cases was still over 96.9% (Table 1b). The average concentrations in these stages were 0.14 and 0.19 mg $L^{-1}$ in GP and 0.06 and 0.02 mg $L^{-1}$ in S column, respectively.

*3.2. Total Nitrogen ($N_{tot}$)*

The total N concentrations in influent water varied during Exp. 1 from 30 to 400 mg $L^{-1}$ and in Exp. 2 from 50 to 175 mg $L^{-1}$.

The total N concentrations in effluents were 10–120 mg $L^{-1}$ during Exp. 1 (Table 1a). Often there were no N-reductions if the N concentrations of influents were very low (less than 80 mg $L^{-1}$). $N_{tot}$-reductions were higher than the required 30% [50] but not in columns B and G at the first normal loading (no reduction found) and high loading stages (B 5.5% and G 24.5% on average), and all columns were below 30% after the normal loading following the low loading stage (no reduction in S and B, and average 4.8% in G) of Exp. 1 (Table 1a). The only statistical difference between the filters was at the high loading stage, when the effluent from column B ($p < 0.05$) contained more $N_{tot}$ (average 107 mg $L^{-1}$) than that from the sand column (average 27 mg $L^{-1}$).

The concentrations of $N_{tot}$ were 2–74 mg $L^{-1}$ during Exp. 2, and all these reductions were clearly higher than the required 30% (Table 1b) [50]. All filters performed equally

at most stages, and only the GP column (average 54 mg L$^{-1}$) at the PO$_4$-P adding stage of normal loading released more N$_{tot}$ ($p < 0.05$) than the S column (average 36 mg L$^{-1}$) (Table 1b).

### 3.3. Chemical Oxygen Demand (COD$_{Cr}$)

The COD$_{Cr}$ of the influent varied highly in both experiments—in Exp. 1 from 60 to 850 mg L$^{-1}$ and in Exp. 2 from 113 to 940 mg L$^{-1}$ (Table 1a,b).

The COD$_{Cr}$ varied from 6 to 152 mg L$^{-1}$ in the effluents of Exp. 1 (Table 1a). The average reductions in the columns S and G were above the 75% given for COD [51] until the end of the low loading stage. Thereafter, the reductions stayed between 60 and 70%. Column B reached 75% for all stages except for the low loading stage (average reduction 51.3%). At the final normal loading stage, column B achieved a higher reduction (on average 87.7%) than column S (on average 62.2%) ($p < 0.05$) (Table 1a).

In Exp. 2, the COD$_{Cr}$ of effluents varied from 36 to 175 mg L$^{-1}$ (Table 1b). The average reductions reached 75% at the second (all columns) and third normal loading stages (S and G) when the influent had high COD$_{Cr}$ (700 and 550 mg L$^{-1}$, respectively) (Table 1b). At other stages, where the influent COD$_{Cr}$ was between 160 and 280 mg L$^{-1}$, none of the columns reached a 75% reduction. The only statistically significant difference ($p < 0.05$) between the filters was at the high loading stage when the GP column (average 50 mg L$^{-1}$) was more efficient than the S column (average 91 mg L$^{-1}$) (Table 1b).

### 3.4. The Electrical Conductivity and the pH in Effluents

The conductivity of influents varied between 700 and 1100 µS cm$^{-1}$ in Exp. 1. The effluent of column S had a lower conductivity (650–1600 µS cm$^{-1}$) than those of the columns B (1000–1600 µS cm$^{-1}$) and G (2000–2700 µS cm$^{-1}$). The conductivity of the effluents of columns B and G were statistically significantly higher than in the effluent of column S. The difference was noticed in column G during the whole experiment ($p < 0.05$): for the first normal load (on average 2530 µS cm$^{-1}$ in column G and 1560 µS cm$^{-1}$ in column S) and the overloading period (on average 2415 µS cm$^{-1}$ in column G and 1530 µS cm$^{-1}$ in column S) ($p < 0.001$) and from low loading (on average 2340 µS cm$^{-1}$ in column G and 750 µS cm$^{-1}$ in column S) to the end of the experiment (on average in each stage 2480–2540 µS cm$^{-1}$ in column G and 720–760 µS cm$^{-1}$ in column S). The difference between column B and column S was significant from the low loading stage until the end of the experiment (at low loading until the PO$_4$ adding stage $p < 0.05$ (average 1140–1190 µS cm$^{-1}$ in column B and 720–760 µS cm$^{-1}$ in column S) and the last normal load (on average 1110 µS cm$^{-1}$ in column B and 740 µS cm$^{-1}$ in column S) $p < 0.01$).

The conductivities of influents in Exp. 2 were between 800 and 2100 µS cm$^{-1}$ (Figure 3). The effluent of column S again had lower conductivities (700–2300 µS cm$^{-1}$) than columns GP (1000–4000 µS cm$^{-1}$) and GSP (2000–3800 µS cm$^{-1}$). The conductivities of effluents in columns GP and GSP started to increase at the low loading stage. The highest conductivities, on average 3700 µS cm$^{-1}$ (max 4000 µS cm$^{-1}$), were measured in the effluent of column GP at normal loading after the low loading stage ($p < 0.05$). GP had the higher conductivity at the PO$_4$ adding stage (2600 µS cm$^{-1}$) than the control S column (1790 µS cm$^{-1}$) ($p < 0.05$) and at last normal stage (2140 µS cm$^{-1}$) than control S column (830 Sm$^{-1}$) ($p < 0.05$).

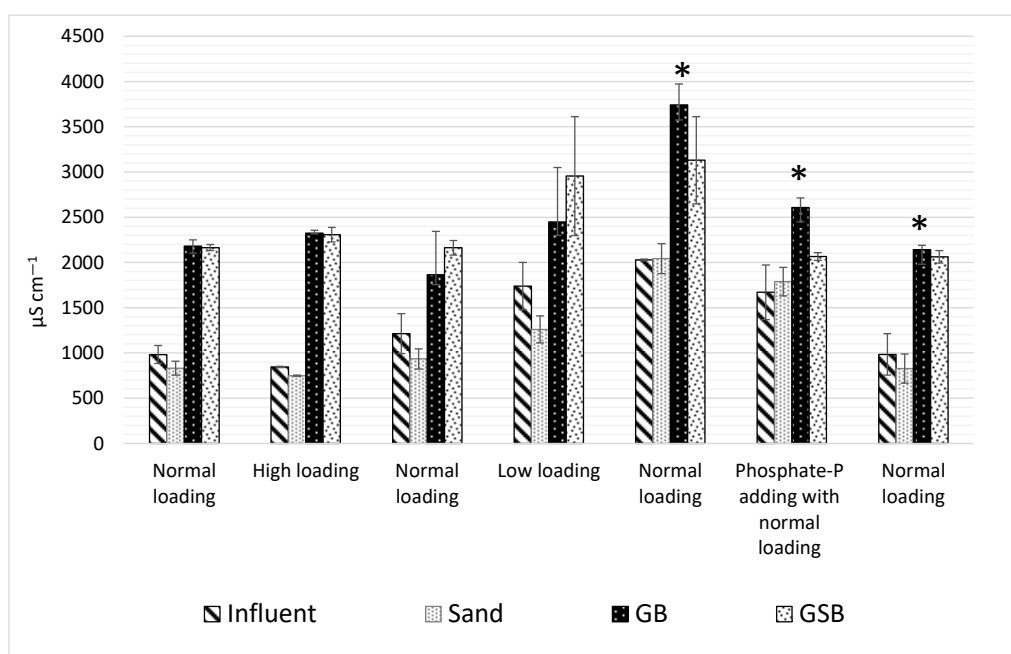

**Figure 3.** The conductivities (μS/cm) of the influents and the effluents from the columns of sand (S); gypsum and peat (GP); and gypsum, sand, and peat (GSP) from Exp 2. Statistically significant differences between the columns S and GP at different loading stages are presented as * = $p < 0.05$. The loading periods are presented in Table 1b.

The pH of all influents and all effluents varied from 5.6 to 8.6. The lowest pH values were measured in the effluents of columns GP ($7.4 \pm 0.7$) and GPS ($7.4 \pm 0.8$). The highest pH was in the B column ($8.1 \pm 0.2$). This result was statistically higher than the pH of the S column ($7.5 \pm 0.5$) from the low loading until the end of the experiment ($p < 0.05$ at a normal load before adding $PO_4$ and at the $PO_4$ adding stage, and $p < 0.01$ at the low loading stage and at normal load after $PO_4$ adding). The pH of the effluent of column G (on average $7.0 \pm 0.7$) was statistically lower than the pH of control S column (on average $7.5 \pm 0.5$) ($p < 0.01$ during the entire experiment, except for $p < 0.05$ at normal loading after high loading, at normal loading after low loading, and at the $PO_4$ adding stage). The only statistical significance in Exp. 2 was during the first normal load when the pH of the GSP (on average 6.5) column was lower than the pH of the sand column (on average 8.3) ($p < 0.05$).

### 3.5. Microbial Quality

The geometric means of microorganisms from influents in Exp. 1 were for fecal coliforms $1.5 \times 10^6$ CFU 100 mL$^{-1}$, intestinal enterococci $1.1 \times 10^5$ CFU 100 mL$^{-1}$, sulfite-reducing clostridia $1.4 \times 10^5$ CFU 100 mL$^{-1}$, somatic coliphages $1.0 \times 10^5$ CFU 100 mL$^{-1}$, and F-specific coliphages $8.1 \times 10^4$ CFU 100 mL$^{-1}$.

All numbers of enteric microorganisms from all effluents in Exp. 1 were below the detection limits (1 CFU 10 mL$^{-1}$, 1 CFU 100 mL$^{-1}$, or 1 CFU mL$^{-1}$), except occasionally for both coliphages 1–9 PFU mL$^{-1}$ in all the columns. The $\log_{10}$-reductions of bacteria and viruses were thus at least 4 to 5 Log$_{10}$-units.

The geometric mean of influent in the Exp. 2 was $1.2 \times 10^6$ CFU 100 mL$^{-1}$ for *E. coli*. The numbers of *E. coli* in the effluents of all the columns were between 1 and 19 CFU/100 mL, without any statistical differences between the columns. Mean reductions of *E. coli* reached 6 $\log_{10}$ for all the columns.

### 3.6. Filter Masses

The $PO_4$-P concentration in the filter mass of column S (sum of the three layers 5 mg kg$^{-1}$) was lower than in the masses of columns B (45 mg kg$^{-1}$ dry weigh) ($p < 0.000$) and G (sum

of layers 98 mg kg$^{-1}$) ($p < 0.000$) (Figure 4.). The middle layer of the column G adsorbed the highest amount of PO$_4$-P, above 70 mg kg$^{-1}$ dry weight (DW).

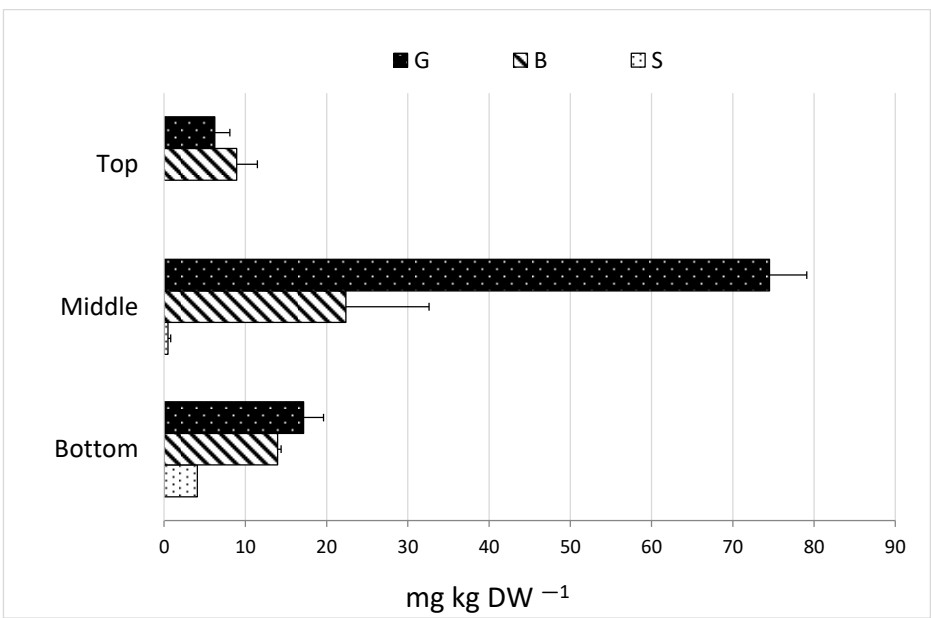

**Figure 4.** PO$_4$-P concentrations (mg in kg DW) ka +/− SD in different column layers of sand (S), biotite (B), and gypsum (G) (Exp. 1). Notice that the middle layer contained the most added biotite or gypsum, respectively.

The N$_{tot}$ concentration in the filter mass of the S column (sum of layers 25 mg kg$^{-1}$ dry weight) was lower than in the masses of columns B (sum of layers 27 mg kg$^{-1}$) or G (sum of layers 30 mg kg$^{-1}$ dry weight) but without statistical significance (Figure 5). Nitrate nitrogen was more abundant in column B (9 mg kg$^{-1}$ dry weight) than in columns S (2 mg kg$^{-1}$ dry weight) ($p < 0.000$) or G (6 mg kg$^{-1}$) ($p < 0.000$), while the ammonium-N form was more abundant in the S (7 mg kg$^{-1}$ dry weight) filter than in the B (0.3 mg kg$^{-1}$ dry weight) filter ($p < 0.01$).

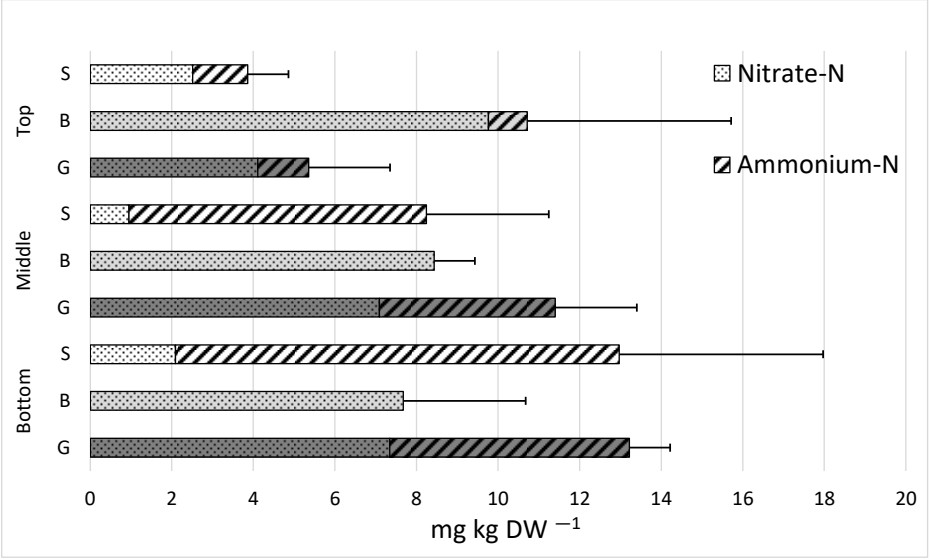

**Figure 5.** Mean NH$_4$-N and NO$_3$-N concentrations as mg per kg of dry weight (DW) in different layers of the sand (S), biotite (B), and gypsum (G) (Exp. 1) and the standard deviation of the sum of ammonium and nitrate nitrogen concentrations.

The numbers of enteric microorganisms in the filter masses were low, less than 80 CFU $g^{-1}$ for all tested bacteria and less than 10 $PFUg^{-1}$ for both phages, without statistical difference between the filter masses. The sand and gypsum column had an average pH of 5.5, while the biotite column had a pH of about 7.

## 4. Discussion

Sand filter systems are a suitable and economically friendly method system for domestic wastewater treatment because of, e.g., low costs and no need for electricity [1–6]. According to The Finnish Government Decree on Domestic Wastewaters in Areas Outside Sewer Networks (517/2017), the purification efficiency in these types of systems must be for total phosphorus ($P_{tot}$) 70%, for total nitrogen ($N_{tot}$) 30%, and for $BOD_7$ 80% [50]. The columns tested in this study generally met these requirements, but the changing conditions caused some reductions in the removal efficiency. Regarding the hygienic quality of the water, the effluents of all columns in all changing conditions fulfilled the hygiene requirements set for irrigation and bathing water [52,53], which is in line with the previous studies [30,31]. Moreover, used masses seem to have reuse potential on the basis of nutrients and hygiene quality.

The effluents of the biotite column met the requirements for different parameters, with a few exceptions for total nitrogen. The effluent also met the requirements of conductivity for irrigation water [52]. Similarly, the effluent of the B column and all the other columns met hygiene requirements for irrigation and bathing water [52,53], which is in line with the previous studies [30,31]. Occasionally, however, private houses' wastewater effluents have caused microbial contamination of groundwaters or waters leaching towards groundwaters [14–18]. Previous results are from real-scale sites (which are older and most often full-scale filters) or pilot-experiments on a larger scale than the current experiment, which may contribute to the difference in results. The effluent passing through the biotite column was slightly alkaline (average pH 8.1). This may have partly converted ammonium to ammonia because alkaline conditions in aqueous solutions favor the ammonia form instead of ammonium. Ammonia may have evaporated from the filter but could have reduced the survival of enteric microorganisms in the filter mass [54,55].

In this work, the used biotite performed well in the varying loading stages but did not improve phosphate removal, although the biotite filter (B) has been found to improve P removal when compared to the ordinary sand filter. Earlier studies with varying setups and wastewater have shown either improved [14,31] or at least similar [15] performance with the ordinary sand filters. The differences between biotite and sand systems in reducing nutrients, etc., may be due to the heterogeneity of filter systems studied and preferential flow. In the previous study, preferential flow and differences of materials have been observed to be a reason for difference between the same type columns to bind phosphorus [56]. Preferential flow can occur, for example, during the construction phase of the filters when the filter materials are packed into place, which may be part of the reason for the current results. Biotite has a lower water permeability [14], which highlights the possible overloading situation, especially in a smaller system.

The use of biotite was also equal with the ordinary sand filter in terms of COD and total nitrogen removal, which is in line with previous findings where no difference between materials with or without biotite was noted [15]. Different types of biotite were used in earlier studies [30,31] which may contribute to the results obtained. Phosphate can interact with the Al and Fe sites in biotite to form a surface complex or precipitate, or cations can bind to biotite upon cation exchange [57,58]. A previous study found that added $CaCl_2$ treatment could improve the P removal capacity of zeolite (which contained biotite) compared to untreated natural zeolites [59]. However, natural zeolites have so far rarely been used because the negative charge on their surface may not improve the removal of phosphates and other anions [59]. Calcium pretreatment of biotite could have improved its ability to remove phosphorus, as previously observed in a zeolite study [59]. Our earlier study [15] has indicated that the removal capacity of a biotite filter for $P_{tot}$, $N_{tot}$, $BOD_7$,

and $COD_{Cr}$ may stay good, also, in the aged filters, in contrast to the ordinary sand filters. There was also no seasonal effect found in the removal efficiency in B filters [15], which supports the results now obtained at cold temperatures (4 °C).

Montmorillonites, to which, e.g., bentonite belongs [60], are widely used in a variety solution including wastewater applications due to their versatile properties [61,62]. Montmorillonites include hexagonal plates between two tetrahedral silica plates [61], where Na+ are hydrated between montmorillonite sheets, while biotite (used in the present work) includes K+ between silica plates. Thus, montmorillonite can swell, and its water permeability is very low; however, biotite does not swell, and, thus, it has a high water permeability that allows the filtration of wastewater [63]. We selected biotite, also, because it is a local byproduct while montmorillonites should be acquired from abroad.

The column using gypsum (G) in Exp. 1 reached most of the chemical requirements (Table 1a), but the performance deviated from that of the S column, especially in phosphate removal. In this study gypsum alone was inadequate to purify a high load of phosphate. Filtra P and Polonite®, which, similarly to gypsum, contain calcium, have earlier been shown to remove >95% phosphorus in a column test with synthetic water [21]. Lower water pH has been noted to decrease the removal of P with Ca-based Polonite filters, but in contrast with C- based Rockfos®, even <9 pH P-$PO_4^{3-}$ was effectively removed when the temperature was above 10 °C [28,64]. In this study, the pH of both influent and effluent from the G column was clearly <9 all the time, and, still, phosphorus removal was efficient at 4 °C except for during the effect of phosphorus addition. Previously high electrical conductivity (EC) values in effluents of soil-based systems have been found to inhibit the removal capacity of P sand filters [56], but this could not be verified here nor whether pH or EC was the reason for the poorer performance of G column.

However, the most important cautionary observation was the high electrical conductivity of effluents of all the gypsum columns, which were at least twice those of the sand columns. For irrigation water, that would mean a severe risk limit. A limit for conductivity of 3000 µScm$^{-1}$ has been set by the FAO for irrigation water [52,65], and this value was exceeded often in the G containing the columns.

In Exp. 2, gypsum was mixed with peat (GP) and sand and peat (GSP). Both these columns met the removal requirements for $N_{tot}$ and $P_{tot}$ ($PO_4$-P), and no drop in reduction of tot-P was observed in contrast to the overload of P in Exp. 1 with gypsum alone (G). However, the P concentration of influent was lower than in the test without peat in Exp 1. For COD, the removal efficiency of the columns GP and GSP was lower than those of the columns of sand—except, for both, at high loading stage and, for GSP, at normal loading after low loading and at $PO_4$ adding stage (Table 1b)

The effluent passing through the biotite column was slightly alkaline (average pH 8.1). This may have partly converted ammonium to ammonia, and the evaporation of ammonia from the B column could also have led to efficient nitrification, as suggested by high concentrations of nitrate and low concentrations of ammonium in the masses of the biotite columns. The masses of all columns contained extractable phosphorus. Nitrogen binding to filter materials was weaker than phosphorus binding, which is in line with the previous real-scale results [66]. It should be noted that the filtration time was only a few months and much less than in a real situation where a sand filter would be used. It is assumed that the age period of use would be about 30 years [67], so the concentration can still be expected to increase under normal conditions. Filtralite®P has been found to accumulate only a little phosphorus in a real-scale sand filter over a 1–2-year period, while in a laboratory experiment where it was saturated with $NaH_2PO_4$ solution, its ability to bind phosphorus was up to 7500 mg P kg$^{-1}$ [68]. This may be due to the difference between influent and $NaH_2PO_4$ solution; it has previously been found that water quality and even slight differences in the filter material affect nutrient removal and binding [57]. This means that, with different influent, the filters used now might have accumulated more nutrients than were now bound to them, and, also, for phosphate, comparing the top layers of the columns shows differences even though they all contained the same sand.

The hygienic quality of all filter masses reached the required level of the Decree of the Ministry of Agriculture and Forestry on Fertilizer Products 2019/1009 [69] with *Escherichia coli* (<1000 cfu g$^{-1}$). Furthermore, the numbers of other tested enteric microorganisms were low. The result is in line with the previous study, where masses used for soil filtration have been found to meet the requirements for bacteria and parasites [69]. It is notable that the columns were fed for only a few months compared to a real situation. A longer time can cause an increase in microbial densities, which was found in another study, where masses contained harmful microbes immediately after decommissioning (had been in use for several years), but after one year of decommissioning, the microbial levels in the masses dropped to a fraction of what they had been immediately after [70]. However, overall research data on decommissioned masses are scarce. The Decree of the Ministry of Agriculture and Forestry on Fertilizer Products [69] requires for *Salmonella* to be analyzed (0/25 g), which was not conducted. The reduction of salmonella has been shown to be equal to that of *E. coli* in a wetland [71], so it can also be assumed that salmonella reduction was in the order of 5 log. The increase of ammonium in the filter material may have improved hygiene as found [72].

Heavy metals could not now be studied here, but previous research has found that they are not a problem for the reuse of masses [68]. It is still difficult to provide general guidance on the beneficial use of mass in all situations [66]; it is necessary to be sure that masses do not affect secondary pollution of the environment [38].

A very high phosphorus concentration in wastewater was studied because, if the number of people producing wastewater is higher, the phosphorus load will increase, especially in arid areas where people must save water. In addition, if people consume larger amounts of milk, milk products, and meat, the wastewater can be richer in phosphorus. Rising wealth in developing countries may also increase the phosphorus load in wastewater through the consumption of more animal-based food.

We have shown earlier that $COD_{Cr}$ has a statistically highly significant positive correlation with $BOD_7$ in domestic wastewater [15]. It has also been shown that $BOD_5$ and COD reductions are nearly identical and linearly correlated [73]. For these reasons and because $COD_{Cr}$ can be determined faster than BOD, the present study used $COD_{Cr}$ to measure the content of organic matter and its reduction. The Decree on Domestic Wastewaters in Areas Outside Sewer Networks (517/2017) [50] do not give any regulation for COD. The obtained results were, therefore, compared with the 75% removal requirement of $COD_{Cr}$ set by Urban Waste Water Directive 91/271/EEC [74].

Organic matter and suspended solids (SS) in the wastewater could cause filter clogging when microorganisms accumulate in biofilms and thicken the surface (near the absorption pipe). Clogging would prolong the water flow, reduce the effective surface area available for water [75], and gradually increase the air flow resistance and create an anaerobic condition [76,77]. This would be harmful for biochemical oxidation and nitrification of ammonium to nitrate [76]. The soil filter is normally used continuously for up to decades. In our earlier study with real, on-site sand filters [15], the performance of some sand filters decreased over time, which may partly be because of clogging. The sand filters used gravity ventilation, as is usually the case. Mechanical ventilation could be more effective, and another study [77] has shown that mechanical ventilation of soil filters effectively increased the removal of $BOD_5$ and $P_{tot}$. In this study, the columns were ventilated by gravity, and no separate ventilation tubes were used in the inner parts of the column. No clogging was observed during this rather short study period.

Sorption processes are reversible, leading to the possibility of P release, which has been observed to occur in a full-scale, experimental, constructed wetland with a calcite-based filter. There, P release was observed when the system was loaded with low P wastewater [78]. In this experiment, a reduction in P removal was observed at normal the load stage after the $PO_4$ adding in the biotite and gypsum filters. In the column filters, bypass flow between the pipe wall and filter mass could cause higher contents in the effluents, reducing the reductions. This could not be verified because the column was

not made of transparent material. It is assumed that bypass flow did not happen, since reduction was not noticed in other parameters but P. The reduction was probably caused by other reasons—perhaps, also, by the release of the accumulated P from the filter mass.

## 5. Conclusions

All the columns were generally quite similar in terms of treatment efficiency, with high microbial reductions and low phosphate and total nitrogen concentrations in their effluent, mostly reaching the treatment levels required by The Finnish Government Decree (517/2017) [50]. The phosphorus binding materials tested in this work did not improve the effluent quality compared to an ordinary sand filter. Waste gypsum can cause high electrical conductivity of the effluent and cannot be recommended as a phosphorus adsorbent.

The experiments were carried out at 4 °C and at varying water flow and phosphorus loading conditions. The results indicate that the temperature or the varying circumstances were not a problem for the function of the soil filters in cold climate areas. In terms of the hygienic quality, the masses are suitable for soil improvement. Soil filtration offers a cheap and simple solution for wastewater treatment.

**Author Contributions:** Conceptualization, K.M., A.-M.V., E.T. and H.H.-T.; methodology, K.M.; software, K.M.; validation, K.M., A.-M.V., E.T. and H.H.-T.; formal analysis, K.M., A.-M.V., E.T. and H.H.-T.; investigation, K.M.; resources, K.M.; data curation, K.M.; writing—original draft preparation, K.M.; writing—review and editing, K.M., A.-M.V., E.T. and H.H.-T.; visualization, K.M.; supervision, A.-M.V., E.T. and H.H.-T.; project administration, K.M.; funding acquisition, K.M. All authors have read and agreed to the published version of the manuscript.

**Funding:** This research was funded by the Finnish Cultural Foundation and Maa- ja Vesitekniikan tuki ry.

**Institutional Review Board Statement:** Not applicable.

**Informed Consent Statement:** Not applicable.

**Data Availability Statement:** Not applicable.

**Acknowledgments:** Doctoral Programme in Environmental Physics, Health, and Biology made it possible to work on the doctoral thesis.

**Conflicts of Interest:** The authors declare no conflict of interest.

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
