# Peer review of "Treatment of Domestic Wastewater in Small-Scale Sand Filters Fortified with Gypsum, Biotite, and Peat"

_sustainability, doi:10.3390/su15021351_

Round 1
Reviewer 1 Report
The entitled paper “Treatment of domestic wastewater in small-scale sand filters 2 fortified with gypsum, biotite, and peat” can be acceptable for publication in the Journal of Sustainability after addressing the following point:
1- Introduction section should be modified and reflect clear enough background.
2- The recycle ability of sand filtration should be studied.
3- The result section should be modified with more details.
4- Using clay materials in different application should be cited in this paper. (-Fuel 320, 123933, 2022 - The Journal of Physical Chemistry C 122 (29), 16498-16509, 2018)
Author Response
REVIEWER 1
RESPONSES TO THE COMMENTS
We thank for the constructive criticism of our manuscript entitled “Treatment of domestic wastewater in small-scale sand filters fortified with gypsum, biotite, and peat.” We have revised the manuscript according to the comments and will give the detailed responses below.
-Introduction section should be modified and reflect clear enough background.
Thank you very much for your criticism regarding the introduction section! We have edited it to better represent the background and the need for research.
-The recycle ability of sand filtration should be studied.
Thank you very much for your comments on the recycle ability of sand filtration! We agree with your suggestion but unfortunately, we are no longer able to continue laboratory research for this article, since it is no longer financially or otherwise feasible. The results gave information on the reuse potential of the masses and we used term “potential” when telling about the results.
-The result section should be modified with more details.
Thank you very much for your criticism on results section. We have edited it and added the details you requested.
-Using clay materials in different application should be cited in this paper. (-Fuel 320, 123933, 2022 - The Journal of Physical Chemistry C 122 (29), 16498-16509, 2018).
Thank you very much for the additional references to part Discussion as you have requested. We have taken them into account when editing the manuscript.
Reviewer 2 Report
In my opinion, the present paper can be accepted after considering to below comments:
1. In my opinion, the introduction is very weak and a literature review should be added based on 2022/2023 research items.
2. Research gap is not determined in the introduction and also, there is insufficient evidence as to whether this constitutes research misconduct.
3. This paper has some English language problems and it should be modified.
4. In the materials and methods section, add a research roadmap.
5. The discussions made in this research are very superficial and should be well compared and deeply evaluated with other articles. In my opinion, this paper needs critical analysis. While, now, it has just some summaries of other studies.
6. Remove the reference from the conclusion. and in the declared section, I did not find the main extract of this article. Please rewrite it.
Author Response
Reviewer 2
RESPONSES TO THE COMMENTS
We thank for the constructive criticism of our manuscript entitled “Treatment of domestic wastewater in small-scale sand filters fortified with gypsum, biotite, and pea.” We have revised the manuscript according to your comments and will give the detailed responses below.
- In my opinion, the introduction is very weak and a literature review should be added based on 2022/2023 research items.
Thank you very much for your criticism regarding the introduction section! We updated it according to your advice, and we added many references based on a request.
- Research gap is not determined in the Introduction and also, there is insufficient evidence as to whether this constitutes research misconduct.
Thank you very much for your criticism. We completed the introduction section to reflect the existing research gap and to reflect better the need for research.
- This paper has some English language problems and it should be modified.
Thank you very much for criticism! The manuscript has been read and corrected by a native English speaker.
- In the materials and methods section, add a research roadmap.
Thank you very much for your request! We added the section (roadmap) you requested to the manuscript.
- The discussions made in this research are very superficial and should be well compared and deeply evaluated with other articles. In my opinion, this paper needs critical analysis. While, now, it has just some summaries of other studies.
Thank you very much for your criticism. We modified the discussion section of the manuscript to compare the results more critically with previous research works.
- Remove the reference from the conclusion. and in the declared section, I did not find the main extract of this article. Please rewrite it.
Thank you very much for your criticism! We did changes according your comment.
Reviewer 3 Report
Dear Editor,
I have some suggestions for the article below:
Although it is said that it is a very up-to-date study, it is necessary to specify where the materials used in the research were obtained or obtained.
Sincerely.
SMT
Author Response
Reviewer 3
RESPONSES TO THE COMMENTS
We thank for the constructive criticism of our manuscript entitled “Treatment of domestic wastewater in small-scale sand filters fortified with gypsum, biotite, and pea.” We have revised the manuscript according to the comments and will give the detailed responses below.
Dear Editor,
I have some suggestions for the article below:
Although it is said that it is a very up-to-date study, it is necessary to specify where the materials used in the research were obtained or obtained.
Sincerely.
SMT
Thank you very much for your comment! We edited the manuscript so that it now contains information about the materials used in the study in Materials and Methods: “Biotite and waste gypsum are by-products from the apatite mine (Yara Finland) from Siilinjärvi Finland. The sand was from Rudus Oy from Kuopio Finland.”.
Round 2
Reviewer 2 Report
I have reevaluated this paper and in my opinion, it is enhanced strongly. Thus, in my opinion, it is suitable for publication.